# Risk-Tailoring Radiotherapy for Endometrial Cancer: A Narrative Review

**DOI:** 10.3390/cancers16071346

**Published:** 2024-03-29

**Authors:** Kristin Hsieh, Julie R. Bloom, Daniel R. Dickstein, Anuja Shah, Catherine Yu, Anthony D. Nehlsen, Lucas Resende Salgado, Vishal Gupta, Manjeet Chadha, Kunal K. Sindhu

**Affiliations:** Department of Radiation Oncology, Icahn School of Medicine at Mount Sinai, New York, NY 10029, USA

**Keywords:** endometrial cancer, uterine malignancy, radiotherapy, radiation oncology, risk-tailoring, personalized treatment, radiation-induced toxicities

## Abstract

**Simple Summary:**

Endometrial cancer is the most common cancer of the female reproductive system in the United States and the second most common cancer of the female reproductive system worldwide. Treatment typically consists of upfront surgery. Postoperative therapy may include adjuvant radiation therapy and/or systemic therapy based on unique patient and pathologic characteristics. In this review, we explore the ways in which patient and tumor characteristics, current and emerging radiation technologies, and other cancer-directed treatments may be considered in developing personalized radiotherapy regimens for patients with endometrial cancer.

**Abstract:**

Endometrial cancer is the most common gynecologic cancer in the United States and it contributes to the second most gynecologic cancer-related deaths. With upfront surgery, the specific characteristics of both the patient and tumor allow for risk-tailored treatment algorithms including adjuvant radiotherapy and systemic therapy. In this narrative review, we discuss the current radiation treatment paradigm for endometrial cancer with an emphasis on various radiotherapy modalities, techniques, and dosing regimens. We then elaborate on how to tailor radiotherapy treatment courses in combination with other cancer-directed treatments, including chemotherapy and immunotherapy. In conclusion, this review summarizes ongoing research that aims to further individualize radiotherapy regimens for individuals in an attempt to improve patient outcomes.

## 1. Introduction

Endometrial cancer is the most common gynecologic cancer in the United States and the second most common gynecologic cancer worldwide [1]. While the incidence of endometrial cancer has been increasing in recent years, overall survival has not improved substantially [2,3]. Endometrial cancer most often presents in post-menopausal women, with greater than 90% of cases being diagnosed after the age of 50 [4,5]. Risk factors for the development of endometrial cancer include older age and chronic exposure to unopposed estrogen [6,7]. Estrogen excess from both exogenous and endogenous sources is currently a leading hypothesis in explaining the recent increase in the incidence of endometrial cancer [8]. Obesity is also thought to increase the risk of developing endometrial cancer via the conversion of androgens to estrogen in adipose tissue. As such, weight loss and physical activity have been proposed as risk-reducing interventions [9,10]. Most endometrial cancer cases are sporadic but about 3% of the cases occur in women with Lynch syndrome [11,12,13].

Although 69% of patients with uterine cancer, of which endometrial cancer is a subtype, are diagnosed with localized disease, the 5-year overall survival rate for such patients varies from 18% for patients with distant disease to 95% for patients with localized disease [3]. Overall survival also varies significantly among individuals from different racial backgrounds. Black women, for example, are more likely to be diagnosed with advanced-stage disease and less likely to survive at any stage or with any histologic subtype of uterine cancer as compared to their White counterparts, contributing to the second largest Black–White survival difference among cancer types in the United States [3,14]. In long-term survivors, quality of life remains a top concern, with menopausal-like symptoms and female sexual dysfunction representing the most common long-term sequelae [15]. Additionally, unlike many other common cancers, endometrial cancer has been marked by a relative absence of major treatment advances, significant racial disparities in terms of diagnosis and survival, and heterogeneous treatment outcomes likely due to a combination of biologic, socioeconomic, and care-related factors that together highlight the need for personalized treatment according to patient and tumor characteristics [3,16,17,18,19].

Radiotherapy plays an essential role in the management of endometrial cancer. In this narrative review, we summarize the current understanding of risk-tailored radiotherapy that is guided by patterns of failure, especially in light of the updated 2023 FIGO staging system. We then discuss how the characteristics of patients and their tumors allow for the tailoring of radiotherapy courses by modality, technique, and regimen. Lastly, we highlight ongoing research that may one day allow for an even more individualized approach.

## 2. Disease Staging in the Molecular Genomic Era

The International Federation of Gynecology and Obstetrics (FIGO) first established a surgical–pathological staging system in 1988; prior to this, they had published clinical staging guidelines for endometrial cancer [20,21]. Many of the fundamental clinical trials for endometrial cancer, such as Gynecological Oncology Group (GOG)-99, Post Operative Radiation Therapy in Endometrial Carcinoma (PORTEC)-1, and PORTEC-2, utilized the FIGO 1988 staging system to define their inclusion criteria [22,23,24,25,26]. In 2009, FIGO published an updated staging system [27,28]. Newer clinical trials, such as GOG-249, GOG-258, and PORTEC-3, utilized the FIGO 2009 staging system [29,30,31]. The aforementioned practice-informing trials, which used either the FIGO 1988 or FIGO 2009 staging system, are referenced in this review. In August 2023, FIGO again updated its staging system to incorporate molecular classifications and histology [32]. As a result, there are new nuances in the management of endometrial cancer in the adjuvant setting, with molecular classifications in addition to traditional tumor characteristics more strongly influencing the selection of treatment. Table 1 compares the FIGO 2009 and 2023 staging systems, with the management tailored to the FIGO 2023 staging system.

Advances in molecular genomic assays have improved our ability to better prognosticate endometrial cancer patients [38]. Molecular classifications are increasingly being used to guide the management of patients with endometrial cancer. In one of the first reports studying outcomes based on molecular classification, Bosse et al. retrospectively assessed patients with FIGO-2009 stage I–IV grade 3 endometrial endometrioid carcinomas. In a study published in 2018, they noted that patients with *POLEmut* and p53abnormal(abn) disease had the best and worst 5-year overall survival and recurrence-free survival rates, respectively [39]. In 2020, León-Castillo et al. published data from patients in PORTEC-3 with high-risk disease, finding that adjuvant chemoradiotherapy improved recurrence-free survival as compared to radiotherapy alone for patients with p53abn disease but outcomes were similarly excellent between the two groups in patients with *POLEmut* disease [37]. Most recently, Horeweg et al. also noted a prognostic significance of *POLEmut* and p53abn in patients enrolled in PORTEC-1 and 2, who mostly had FIGO-2009 stage I disease [35]. As such, the authors suggested that radiotherapy may be omitted in patients with *POLEmut* disease, while external beam radiotherapy (EBRT) (in lieu of vaginal brachytherapy [VBT] or no adjuvant therapy) should be given in patients with p53abn disease. As a result of this work on molecular markers for endometrial cancer, ESGO/ESTRO/ESP published an updated management guideline in 2021 that encourages the use of novel molecular classification for all patients with endometrial carcinomas and integrates the molecular markers, including POLEmut, MMRd/NSMP, and p53abn, into the definition of prognostic risk groups [13]. The updated FIGO 2023 staging system further encourages molecular classification testing, including for POLEmut, MMRd, NSMP, and p53abn, and incorporates the presence of *POLEmut* and p53abn into the FIGO stage I and II classification [32]. The presence of *POLEmut* and p53abn does not modify the FIGO stage III and IV classification, while the status of MMRd and NSMP does not modify FIGO staging at all.

## 3. Risk-Tailoring Radiotherapy by Pathologic Risk Factors and Patient Characteristics

Per the National Comprehensive Cancer Network (NCCN), the standard of care for patients with medically operable cases of endometrial cancer starts with a total hysterectomy and bilateral salpingo-oophorectomy (TH/BSO) along with an assessment of regional lymph nodes [33]. For patients who are eligible for and whose goals of care include fertility-sparing management, definitive oncologic resection and any adjuvant radiotherapy may be forgone but these patients must be counseled that fertility-sparing management is not the standard of care treatment [33]. For select patients with stage IA disease who lack adverse risk factors, observation may be pursued after definitive oncologic resection [40]. Adjuvant therapy, in the form of radiotherapy with or without systemic therapy, is commonly offered to those with more advanced disease or adverse clinical or pathologic features to minimize the risk of disease recurrence. 

The decision to offer adjuvant radiotherapy is influenced by several patient-related factors, including age and performance status. The age-associated recommendation for early-stage endometrial cancer has been established through several clinical trials. In PORTEC-1, age > 60 years old, grade 3 disease, and deep (≥50%) myometrial invasion were shown to be unfavorable prognostic factors for patients with FIGO Stage IB and IC disease (per FIGO 1988 staging system) [22,23]. Thus, patients who are older than 60 years and have at least one additional risk factor are classified as having high intermediate-risk disease; adjuvant radiotherapy is recommended over observation in this patient population. Age was also used as a criterion for the high intermediate-risk group in GOG 99, which examined patients with FIGO Stage IB, IC, and II (occult) disease (per FIGO 1988 staging system) [24]. Specifically, patients who were 50 years of age and older and had at least two risk factors (grade 2 or 3 disease, lymphovascular invasion, or invasion of the outer 1/3 of the myometrium) or 70 years of age and older and at least one risk factor were defined as having high intermediate-risk disease. GOG 99 showed a statistically significant reduction in the incidence of locoregional recurrence with adjuvant radiotherapy as compared to observation in this population. 

Taken together, patients with early stage endometrial cancer are stratified by adverse clinical and pathologic risk factors including age, pathologic grade, histologic subtype, depth of myometrial invasion, and lymphovascular space invasion (LVSI). In general, a more aggressive approach to adjuvant therapy, such as adjuvant radiotherapy over observation, is considered in patients with numerous risk features. Table 1 elaborates on how certain tumor characteristics may affect management. In contrast, for patients with more advanced disease, adjuvant radiotherapy (specifically EBRT with or without VBT) is typically warranted, with systemic therapy being considered and sometimes offered depending on the patients’ unique pathologic features and risk for locoregional and metastatic disease. Both the PORTEC and GOG study series have been instrumental in defining the role of adjuvant radiotherapy in the treatment of endometrial cancer, including whether to offer observation, EBRT, VBT, or both EBRT and VBT after surgery. 

Patient characteristics including performance status and ability to tolerate treatment must be considered when determining the most appropriate individualized treatment course. If a patient is not a surgical candidate due to overall health concerns and/or medical comorbidities, definitive radiation therapy or definitive chemoradiotherapy are potential alternatives [33]. In such cases, radiotherapy is tailored to the patient’s anatomy, with radiation delivered to the uterus, cervix, and upper vagina as per the American Brachytherapy Society (ABS) consensus statement for medically inoperable endometrial cancer [41]. Though radiation is well tolerated in patients with medical comorbidities, clinicians should evaluate patients for their medical stability and tolerance to undergo radiotherapy, including the simulation and treatment process. 

## 4. Radiotherapy Technical Considerations and Dose Fractionation

Endometrial cancer is a complex disease site from a radiotherapy standpoint and several treatment modalities may be utilized. Adjuvant EBRT alone or with VBT is favored for high-grade Stage IB and above endometrial cancer per the 2009 FIGO staging system [33]. Per the ASTRO clinical practice guideline for endometrial cancer, a total EBRT dose of 45–50.4 Gy should be delivered [34]. Adjuvant EBRT is optimally delivered using intensity-modulated radiation therapy (IMRT) to facilitate careful delineation of the target and reduce treatment toxicities [42,43,44,45]. As such, contouring atlases are a valuable tool for physicians wishing to treat these patients with this technique [46]. An internal targeting volume (ITV) should be used to account for the motion of the parametrium, vaginal cuff, and paravaginal tissues as a result of changes in bladder filling [46]. In addition, daily image guidance is advised to ensure correct bladder filling and adequate targeting [46]. 

VBT alone has become the standard of care for patients with high-grade FIGO Stage IA as well as low to intermediate-grade FIGO stage IB disease, per the 2009 FIGO staging system [25,26,34,47]. A wide variety of VBT monotherapy dosing and fractionation regimens exist, with the ABS noting 23 separate trials, each with its own VBT prescription, assessing the efficacy of adjuvant VBT monotherapy [48]. Per the 2014 ABS survey, the most common VBT monotherapy prescriptions utilized were 7 Gy × 3 fractions to a depth of 0.5 cm and 6 Gy × 5 fractions to the surface. Additionally, VBT may be employed as a boost after EBRT [34]. Per the ABS consensus guidelines, the goal of EBRT with a VBT boost is to deliver at least a total of 65–70 Gy to the vaginal surface in most patients and up to 70–75 Gy in patients with superficially positive margins [49]. A wide range of VBT boost regimens are also utilized, with the most common being five Gy × 3 fractions to a depth of 0.5 cm and six Gy × 3 fractions to the surface per the 2014 ABS survey [50]. 

While there is variability among practitioners, VBT is most commonly prescribed to the proximal 3–5 cm or 1/3–1/2 of the vagina [48,50]. VBT is delivered using a vaginal cylinder or ovoids. As there is no literature supporting one applicator over the other in terms of local disease control, the choice of the applicator is usually institution-, provider-, and patient-specific [48]. Multichannel vaginal cylinders may also be used. The use of this applicator is optimal when differential loading patterns are required to shape the radiation dose distribution, potentially allowing for a reduction in dose to the bladder and rectum [51,52]. The patient’s postoperative vaginal anatomy, specifically its length and shape, may help guide applicator selection. A vaginal cylinder may be of a predetermined length or composed of segments that can be assembled to the length of interest, thus allowing for treatment of the entire length of the vagina, while ovoids may treat only the upper part of the vagina. Additionally, a cylinder may be preferred for a narrow roughly tubular vagina, while ovoids may be chosen if there are surgical remnants of the vaginal fornices (with a resulting expansion of the lateral apices).

There are a few important considerations in selecting a custom/best-fit applicator for a given patient. The cylinder comes in various sizes with diameters ranging from 2.0–4.0 cm, while the ovoids also have several available sizes [53,54]. The largest diameter fitting the anatomy is typically used. This ensures minimal risk of air gaps between the applicator surface and the vaginal mucosa, allowing a homogenous dose distribution to the target [48]. Single and multichannel cylinders are available. The 2014 ABS survey found that the vaginal cylinder is the most commonly utilized applicator for postoperative endometrial cancer patients, with a majority of providers using a single-channel cylinder with high dose rate (HDR) [50]. CT-based planning is also preferred to ensure accurate dosimetry [50]. 

## 5. Radiotherapy-Related Toxicities and Management

Radiation-related toxicities, including gastrointestinal, genitourinary, skin, bone, and sexual toxicity, in patients with endometrial cancer may develop during or after treatment and may persist for varying lengths of time. Some common acute radiotherapy-related toxicities include fatigue, skin irritation, diarrhea, and dysuria and late toxicities include vaginal shortening, pelvic fractures, and adhesions [55]. For long-term survivors, quality of life, in addition to the oncologic outcome, is an important concern. Thus, attempts to address both acute and late toxicities are crucial to patient care and select studies with quality-of-life endpoints will be highlighted in a later section. The development, severity, and longevity of radiation-related toxicities depend on non-radiation-related factors as well. For example, risk factors that contribute to late gastrointestinal toxicity include diabetes, inflammatory bowel disease, connective tissue disease, HIV, prior abdominal surgery, low body mass index, tobacco use, and concurrent chemotherapy [56]. 

The particular choice of the radiation treatment modality employed plays an important role in reducing the incidence and severity of radiation-related toxicity. For example, as compared to 3DCRT, IMRT was associated with less toxicity and improved quality of life among patients with endometrial cancer in RTOG 1203 [43]. In this study, patients’ gastrointestinal and genitourinary symptoms, such as diarrhea, fecal incontinence, abdominal pain, dysuria, urinary incontinence, and incomplete bladder emptying, were quantified using the bowel and urinary domains of the Expanded Prostate Index Composite (EPIC), respectively [43]. Patients treated with IMRT reported a significantly smaller decline in mean bowel and urinary scores as compared to patients treated with conventional treatment at week 5 (mean bowel score decline: 18.6 versus 23.6 points, *p* = 0.048; mean urinary score decline: 5.6 versus 10.4 points, *p* = 0.03). At three years after treatment, patients treated with IMRT continued to report fewer gastrointestinal and genitourinary symptoms than their counterparts [45]. 

VBT can also affect the degree of toxicities experienced by survivors of endometrial cancer. A randomized trial for patients with high-intermediate risk endometrial cancer, PORTEC-2, showed that patients treated with vaginal cuff brachytherapy experienced fewer toxicities as compared to patients treated with adjuvant EBRT (IMRT or 3DCRT) [25]. Thus, when considering treatment options including radiotherapy, one should look to optimize oncologic outcomes such as disease-free survival while minimizing toxicity and the impact on quality of life.

Patients with endometrial cancer may also suffer from sexual dysfunction, including painful receptive intercourse, difficulties with arousal and orgasm, and decreased libido [57]. Radiation can cause vaginal atrophy and vaginal stenosis, potentially contributing to painful receptive vaginal intercourse [58]. To mitigate vaginal stenosis, vaginal dilation during and after treatment can help preserve vaginal patency [58,59]. Pelvic radiation can also damage the female erectile tissues or bulboclitoris [60]. A pilot study of patients treated with radiation for cervical cancer found that a vacuum clitoral device could potentially help restore sexual pleasure [61]. Radiation can also damage the pelvic floor muscles and lead to anorgasmia or dysorgasmia [62]. Strengthening the pelvic floor may help restore the ability to achieve orgasm and kegel exercises can be implemented as a rehabilitation strategy for endometrial cancer survivors with sexual dysfunction [59,63]. 

Lastly, pelvic radiation for the treatment of endometrial cancer has implications for transgender men or transmasculine people in general, who have undergone or plan to undergo genital affirmation surgery [64]. While there are few guidelines on how to manage gender minority patients with endometrial cancer and manage the impacts of treatment on sexual function, radiation, similar to other cancer-directed therapies, can impact genital affirmation surgery. For example, delivering radiation prior to genital affirmation surgery may cause surgical complications during and following reconstruction. It is thus important to counsel transgender patients with endometrial cancer who are planning on undergoing gender affirmation surgery on the potential complications that radiation treatment might have on the surgery itself [65]. 

## 6. Combining Radiotherapy with Other Adjuvant Cancer-Directed Systemic Treatments

### 6.1. Chemotherapy

In patients with stage III-IV endometrial cancer or high-risk histologies, such as serous, clear cell, mixed histology, undifferentiated, and carcinonsarcoma, chemotherapy may be given alone, concurrently, or sequentially with radiotherapy [33]. Per the NCCN guidelines, cisplatin remains the treatment of choice when chemotherapy is given concurrently with EBRT as a radiosensitizing agent, while carboplatin and paclitaxel are most commonly delivered in the absence of radiotherapy [33]. Several retrospective studies have demonstrated the efficacy of various sequencing approaches, including concurrent chemoradiation followed by chemotherapy (RT + chemotherapy → chemotherapy) [30,31], chemotherapy before and after radiotherapy (the “sandwich” approach; chemotherapy → RT → chemotherapy) [66], and chemotherapy followed by sequential radiotherapy (chemotherapy → RT) [67]. In theory, upfront chemotherapy may be ideal for patients with a higher risk of distant metastasis while upfront chemoradiotherapy may be provided to those with a higher risk of locoregional recurrence. 

### 6.2. Hormone Therapy

Tamoxifen and other hormone therapies may be used in the treatment of endometrial cancer [68]. Specifically, megestrol acetate/tamoxifen and everolimus/letrozole have been utilized for patients with recurrent or metastatic endometrial carcinoma, select non-surgical candidates with uterine-limited disease, and those interested in fertility preservation [33]. There is, however, very limited data on the use of these therapies with radiotherapy. 

### 6.3. Immunotherapy

Chemoimmunotherapy, in the form of carboplatin/paclitaxel combined with either pembrolizumab or dostarlimab-gxly, is a category 1 recommended therapy on the NCCN guidelines for select patients with stage III/IV endometrial carcinoma or recurrent disease [69,70]. There is no published trial data to date that examines the use of combination immunotherapy and radiotherapy for patients with endometrial carcinoma. 

### 6.4. Targeted Therapy

A kinase inhibitor, lenvatinib, plus pembrolizumab is a category 1 recommended therapy per NCCN guidelines for mismatch repair (MMR)-proficient recurrent endometrial carcinoma [33,71]. An mTOR inhibitor, everolimus, and letrozole may be offered to select patients, as mentioned in a prior section. In combination with carboplatin/paclitaxel, bevacizumab, an angiogenesis inhibitor, is a first-line therapy for recurrent disease, while trastuzumab is a first-line therapy for HER-2 positive uterine serous carcinoma or carcinosarcoma [72,73,74]. No published reports to date have examined the use of radiotherapy with targeted therapy in patients with endometrial cancer.

## 7. Post-Treatment Surveillance

Per the NCCN guidelines, regular physical exams, including pelvic exams, are recommended following treatment, while imaging should be ordered as clinically warranted [75]. This recommendation is supported by the TOTEM study, a randomized clinical trial that compared two follow-up regimens, intensive versus minimalist, for patients with endometrial cancer and found an intensive follow-up regimen does not improve overall survival [76]. Continued patient education on sexual health, including the use of a vaginal dilator, and counseling on late treatment effects is additionally recommended [75]. Treatment with or without radiotherapy does not impact the post-treatment surveillance recommendation.

## 8. Future Directions

Improvements in surgical techniques, increasing availability of molecular assays, and advances in radiation technology are stimulating research into the personalization of radiotherapy for patients with endometrial cancer, including studies on molecular features and classification, sexual toxicity, pelvic physical therapy, dose fractionation, and treatment techniques. Table 2 shows a list of ongoing phase III clinical trials involving radiotherapy in patients with endometrial cancer.

### 8.1. Studies with Quality of Life Endpoints

Sexual dysfunction is an understudied aspect of radiotherapy in patients with endometrial cancer. One ongoing phase III trial, NCT03785288, is randomizing patients to two different HDR brachytherapy dose fractionation regimens (seven Gy × 3 fractions versus four Gy × 6 fractions) to determine the optimal brachytherapy schedule as it relates to sexual dysfunction following treatment [77]. The researchers aim to determine which approach is favored by patients, the effects of each fractionation schedule on patient-reported sexual dysfunction, and how sexual function relates to changes in vaginal anatomy. The results of this study may help determine the optimal treatment course for patients requiring VBT following surgery for endometrial cancer. Another study, NCT04544735, is seeking to better understand the factors associated with sexual dysfunction and quality of life in women who have undergone pelvic radiotherapy [84]. The researchers plan to use information gathered from patients, oncologists, and physical therapists to develop a comprehensive physical therapy plan and test these interventions on patients undergoing pelvic radiotherapy. The information generated from this study will be critical in improving our understanding of how pelvic radiotherapy impacts sexual function and quality of life and in determining the best way to utilize appropriate interventions in the clinic. 

Ongoing trials are also assessing different radiotherapy techniques. A phase I trial, NCT04567771, is comparing toxicities and quality of life in patients with endometrial or cervical cancer who are treated with proton beam therapy or photon IMRT [85]. This study will use the EPIC bowel and urinary domains, Common Terminology Criteria for Adverse Events (CTCAE) criteria, and quality of life questionnaires to determine whether proton therapy improves toxicity outcomes in patients with gynecological malignancies who require pelvic radiotherapy as compared to the standard IMRT approach. This study will provide further information regarding the potential benefits of proton therapy in this setting.

### 8.2. Studies with Clinical Endpoints

There is also ongoing research into the utilization of shorter radiation schedules for pelvic radiotherapy in endometrial cancer. Similar studies have been completed in cancers of other disease sites, which have demonstrated similar toxicity profiles and comparable oncologic outcomes between shorter and longer radiation courses [86,87,88,89,90]. Additionally, these shorter courses of treatment have been associated with decreased costs to the healthcare system and improved access and convenience for patients [91]. One study evaluating the safety and efficacy of a hypofractionated pelvic radiotherapy approach in patients with endometrial cancer is RT-PACE [92,93]. This phase I/II trial will first determine the maximally tolerated dose of a 16-fraction IMRT course of pelvic radiotherapy and will then compare toxicity and efficacy endpoints to patients receiving treatment with standard fractionation IMRT on RTOG 1203. Another study, DeCRESCendo, will look to evaluate the safety and efficacy of a five-fraction (25 Gy) pelvic radiotherapy regimen in patients with advanced endometrial adenocarcinoma or with high-risk histology [94]. These studies will allow for the continued individualization of radiotherapy options for patients with endometrial cancer and may increase access to care while lowering treatment costs.

## 9. Conclusions

Endometrial cancer has been increasingly recognized as a heterogeneous disease, though radiotherapy remains a cornerstone of its management. The advent of novel disease classifications and therapeutic approaches has highlighted a need for personalized treatment according to the unique molecular and histologic features of each tumor. There is a plethora of ongoing research aimed at improving outcomes and personalizing radiotherapy for patients with endometrial cancer. These trials, along with other ongoing studies, will hopefully lead to improvements in identifying optimal treatment schedules, reducing treatment burdens on patients in the form of secondary toxicities, and providing information on how to best incorporate emerging technologies into practice.

## Figures and Tables

**Table 1 cancers-16-01346-t001:** Comparison of the FIGO 2009 and FIGO 2023 endometrial cancer staging system with the management tailored to the FIGO 2023 staging system.

Stage	FIGO 2009 [28]	FIGO 2023 [32] (Difference with Prior Staging System in Red)	Management Post-TH/BSO ^†^ (Preferred Management in Bold)
I	Confined to uterine corpus	Confined to uterine corpus + ovary	
IA	<1/2 MI	See below	
IA1	-	Confined to a polyp or endometrium + low-grade endometrioid carcinoma	**Observation** [33,34]
IA2	-	<1/2 MI + low-grade endometrioid carcinoma + no/focal LVSI	**Observation** or VBT (consider if age ≥ 60 and/or LVSI, strongly consider if both factors present) [33]
IA3	-	<1/2 MI + low-grade endometrioid carcinoma + confined to uterus and unilateral ovary without capsule involvement + no/focal LVSI	**Observation** per FIGO guideline [32]
IAm_POLEmut_ ^‡^	-	[Downstaged] *POLEmut* endometrial carcinoma + confined to uterus (cervical extension allowed)	Observation per combined PORTEC-1 and 2 analysis [35] and per meta-analysis [36]
IB	≥1/2 MI	≥1/2 MI + low-grade endometrioid carcinoma + no/focal LVSI	**VBT** [33,34], observation (consider if age < 60 and no LVSI) [33], or EBRT (consider if age ≥ 60 and/or LVSI) [33]
IC	-	Aggressive histology + confined to a polyp or endometrium	**Endometrioid histology: VBT**, observation (consider), or EBRT (consider if age ≥ 70 or LVSI) [33] Non-endometrioid histology: ± EBRT ± VBT ± systemic therapy [33]
II	Invades cervical stroma + confined to uterus	See below	
IIA	-	Invades cervical stroma + low-grade endometrioid carcinoma	**EBRT** and/or VBT ± systemic therapy [33]
IIB	-	Low-grade endometrioid carcinoma + extensive LVSI	**EBRT** and/or VBT ± systemic therapy [33]
IIC	-	Aggressive histology + any MI	EBRT ± VBT ± systemic therapy, or systemic therapy ± EBRT ± VBT [33]
IICm_p53abn_ ^‡^	-	[Upstaged] p53abn + confined to uterus (± cervical invasion) + any MI	EBRT + systemic therapy per exploratory subanalysis of PORTEC-3 [37]
III	Local/regional involvement		Systemic therapy ± EBRT ± VBT [33]
IIIA	Serosa and/or adnexa		
IIIA1	-	Ovary/fallopian tube	
IIIA2	-	Uterine subserosa/serosa	
IIIB	Vagina and/or parametrium	Vagina and/or parametrium OR pelvic peritoneum	
IIIB1	-	Vagina and/or parametrium	
IIIB2	-	Pelvic peritoneum	
IIIC	Pelvic and/or para-aortic LN		
IIIC1	Pelvic LN		
IIIC1i	-	Micrometastasis	
IIIC1ii	-	Macrometastasis	
IIIC2	Para-aortic LN		
IIIC2i	-	Micrometastasis	
IIIC2ii	-	Macrometastasis	
IV	Bladder/bowel invasion and/or distant metastasis		
IVA	Bladder and/or bowel		Systemic therapy ± EBRT ± VBT [33]
IVB	Distant metastasis	Abdominal peritoneal metastasis outside of pelvis	Upfront TH/BSO may not be appropriate Systemic therapy ± EBRT ± VBT [33]
IVC	-	Distant metastasis	

Abbreviations and definitions: Aggressive histology—high-grade (grade 3) endometrioid, serous, clear cell, carcinosarcomas, undifferentiated, mixed, and unusual subtypes. HIR—high intermediate risk. LN—lymph node. Low-grade—grade 1 and 2. LVSI—lymphovascular space involvement. Extensive LVSI—≥5 vessels. Focal LVSI—<5 vessels. MI—myometrial involvement. ^†^ The management is per the NCCN and/or ASTRO guidelines based on the FIGO 2009 staging [33,34]. If there is no available NCCN or ASTRO guideline recommendation, then select publication is cited. ^‡^ The presence of *POLEmut* and p53abn modify the FIGO stage I and II classification.

**Table 2 cancers-16-01346-t002:** Ongoing phase III clinical trials assessing the role of radiotherapy.

NCT Number; Protocol Number	Stage (Per FIGO 2009 Staging System Unless Otherwise Noted); Histology	N; Treatment Arms	Primary Outcome Measures	Sequence of RT	Trial Status
NCT03785288 [77]	Stage I–II (grade 1–3) Endometrioid, serous, clear cells, and carcinosarcoma histologic pathologies	N = 258; HDR VBT 7 Gy × 3 fractions or HDR VBT 4 Gy × 6 fractions (with option for patients to decline their randomization and switch to the other treatment arm)	Female sexual function index and preference option randomized design at 1 year post treatment	RT delivered 4–12 weeks after surgery (TH/BSO with or without lymph node dissection)	Recruiting
NCT03422198 [78,79]	Stage IA, (grade 2 and 3 only), stage IB or stage II (grades 1–3) Endometrial carcinoma (including endometrioid type, serous, and clear cell), carcinosarcoma, and other sarcoma	N = 108; Short course VBT 11 Gy × 2 fractions (1 week apart) at the surface or Standard of care VBT for 3–5 fractions within 3 weeks	Quality of Life (QOL) at 1 month post treatment	RT delivered ≤ 16 weeks after hysterectomy	Recruiting
NCT00006027; RTOG-9905; GOG-0194 [80]	Stage IC–IIB (grade 2–3) (per FIGO 1998 staging system) Endometrioid endometrial adenocarcinoma with <50% papillary serous or clear cell histology	N = 436; RT once daily 5 days a week for 5.5 weeks or RT as above with concurrent IV cisplatin, followed by paclitaxel IV alone after RT completion	(Primary objectives not explicitly stated.) Relapse-free survival; patterns of recurrence; acute and late toxicity profiles	RT or concurrent CRT then CTX alone within 8 weeks after surgery (TH/BSO).	Terminated [81]
NCT04634877; MK-3475-B21; KEYNOTE-B21; ENGOT-en11; GOG-3053 [82]	High recurrence risk disease defined as stage I/II with MI of non-endometrioid histology or any histology with p53 mutation/aberrant expression, or stage III or IVA of any histology. Endometrial carcinoma or carcinosarcoma (mixed Mullerian tumor)	N = 990 IV pembrolizumab each 3-week cycle (Q3W) for 6 cycles followed by IV pembrolizumab each 6-week cycle Q6W for an additional 6 cycles or IV placebo each Q3W for 6 cycles followed by IV placebo Q6W for an additional 6 cycles	Disease-free survival; overall survival	Curative intent surgery that included TH/BSO is first, followed by SoC CTX for 4 or 6 cycles (with optional EBRT and/or VBT starting within 6 weeks of SoC CTX completion) that are given during the Q3W pembrolizumab or placebo period.	Active, not recruiting
NCT04214067; NRG-GY020 [83]	Stage II or stage I with the following combination: Age ≥ 70 with ≥1 risk factors; Age ≥ 50 and <70 with ≥ 2 risk factors; Age < 50 with 3 risk factors. Risk factors: MI ≥ 50%; LVSI; Grade 2 or 3. Endometrioid endometrial cancer	N = 168; EBRT for 5–6 weeks and VBT completed within 7 days after completion of EBRT or RT as above plus IV pembrolizumab Q6W for up to 9 cycles starting within 7 days prior to the start of RT	Recurrence-free survival at 3 year	Surgical staging (including hysterectomy, removal of cervix, bilateral salping-oophorectomy) first, snf then EBRT with VBT.	Active, not recruiting

Abbreviation: CTX—chemotherapy. CRT—chemoradiotherapy. EBRT—external beam radiotherapy. IV—intravenous. LVSI—lymphovascular space invasion. MI—myometrial invasion. NCT—National Clinical Trial. RT—radiotherapy. SoC—standard of care. TH/BSO—total hysterectomy and bilateral salpingo-oophorectomy. VBT—vaginal brachytherapy.

## Data Availability

No new data were created or analyzed in this study. Data sharing is not applicable to this article.

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
