# Peer review of "Risk-Tailoring Radiotherapy for Endometrial Cancer: A Narrative Review"

_cancers, 2024, doi:10.3390/cancers16071346_

Round 1

Reviewer 1 Report

Comments and Suggestions for Authors

GENERAL

ARTICLE TYPE review article

This review article aims to address the history and updates of radiotherapy alone or in combination with other modalities in endometrial cancer, in connection with current investigations and the transitions in FIGO staging systems. Also, the use of individualized radiation method has been discussed to reduce treatment-related morbidity.

The simple summary, abstract, introduction, literature review sections, and conclusion are well arranged and addressed. However, the presentations in Tables should be more condensed and improved. I have some comments listed as the following:

In general

Please add ESGO/ESTRO/ESP guideline in this review, as one of the reference accordingly (updated and published in 2021, rather than 2016). The tables and the text body with its citation should be added and corrected. 

Table 1

1.      Please re-confirm the management of FIGO stage IA3 (2023 FIGO staging), based on ESGO/ESTRO/ESP or NCCN guidelines.

2.      Please improve the use of ‘+/-’ systemic therapy in stage III and IV for more appropriate expression as the recommended modality category based on the evidence.

Table 2.

1.      The column ‘trial name’ could be replaced by ‘Protocol No./NCI trial no.’, for instance, MK-3475-B21; KEYNOTE-B21; ENGOT-en11; GOG-3053; NCT04634877 [79]. The title of the trial could be omitted.

2.       The inclusion participant numbers in each could be stated in each arm, within the column ‘Treatment arms’. The total participant numbers in ‘trial name’ could be omitted.

3.      The primary/secondary end points could be listed, without the column ‘Conclusion or study goals’.

4.      The column ‘Results of primary objectives’ may be replaced by ‘ Trial status’, for example: under recruitment, closed and under analyses, or early closed due to poor accrual, etc. 

Reviewer 2 Report

Comments and Suggestions for Authors

The paper should be interesting but it is not clear which protocol has been adopted to performe this review.

I suggest to report in clear how the references has been selected.

An addition it  is not so clear the aim of this paper or better which kind of add value do you expect from this analysis.

The evaluation of the outcome in term not only of survival but also in toxicity are analyzed very superficially and no information about the follow up regimen used.

Reviewer 3 Report

Comments and Suggestions for Authors

Firstly, I would like to extend my gratitude to the authors for this excellent, highly clinically-oriented review. As a radiation oncologist, I found this comprehensive summary of the current landscape of endometrial cancer to be invaluable. With the advent of new classifications and subsequent biomolecular stratifications, providing guidance for adjuvant treatments has become increasingly complex. The authors' dedicated effort to elucidate a field that remains nebulous and fraught with various grey areas is truly commendable. The tables presented in the article are exceptionally practical and could easily be carried in one's 'gown pocket' for reference at the patient's bedside. My only suggestion would be to ensure that the title clearly reflects the type of review to expect. Regarding the English writing, I have no further comments.

Author Response

Reviewer 3 Comments:

Is the work a significant contribution to the field? 4 stars

Authors’ response: We thank the reviewer for this comment.

Is the work well organized and comprehensively described? 4 stars

Authors’ response: We thank the reviewer for this comment.

Is the work scientifically sound and not misleading? 4 stars

Authors’ response: We thank the reviewer for this comment.

Are there appropriate and adequate references to related and previous work? 4 stars

Authors’ response: We thank the reviewer for this comment.

Is the English used correct and readable? 5 stars

Authors’ response: We thank the reviewer for this comment.

Firstly, I would like to extend my gratitude to the authors for this excellent, highly clinically-oriented review. As a radiation oncologist, I found this comprehensive summary of the current landscape of endometrial cancer to be invaluable. With the advent of new classifications and subsequent biomolecular stratifications, providing guidance for adjuvant treatments has become increasingly complex. The authors' dedicated effort to elucidate a field that remains nebulous and fraught with various grey areas is truly commendable. The tables presented in the article are exceptionally practical and could easily be carried in one's 'gown pocket' for reference at the patient's bedside.

Authors’ response: We greatly appreciate this thoughtful comment.

My only suggestion would be to ensure that the title clearly reflects the type of review to expect.

Authors’ response: We thank the reviewer for this comments. As the reviewer has recommended, we have revised the title to “Risk-Tailoring Radiotherapy for Endometrial Cancer: A Narrative Review.”

Regarding the English writing, I have no further comments.

Authors’ response: We thank the reviewer for this comment.

Reviewer 4 Report

Comments and Suggestions for Authors

Dear Authors,

I found the manuscript very interesting and well written, I have really appreciated the focus on the future directions in term of tailored radiotherapic approach based on patients' and disease's characteristics in order to reduce not only the costs but especially the adverse effects related to treatment without compromise the efficacy of treatment and prognosis of patients affected. 

Comments on the Quality of English Language

Dear Authors,

I think the manuscript is well written, English language needs of only minor editing.

Author Response

Reviewer 4 Comments:

Is the work a significant contribution to the field? 4 stars

Authors’ response: We thank the reviewer for this comment.

Is the work well organized and comprehensively described? 4 stars

Authors’ response: We thank the reviewer for this comment.

Is the work scientifically sound and not misleading? 4 stars

Authors’ response: We thank the reviewer for this comment.

Are there appropriate and adequate references to related and previous work? 4 stars

Authors’ response: We thank the reviewer for this comment.

Is the English used correct and readable? 3 stars

Authors’ response: We hope that the minor language editing we have made to the manuscript will satisfy the reviewer’s concerns.

I found the manuscript very interesting and well written, I have really appreciated the focus on the future directions in term of tailored radiotherapic approach based on patients' and disease's characteristics in order to reduce not only the costs but especially the adverse effects related to treatment without compromise the efficacy of treatment and prognosis of patients affected.

Authors’ response: We greatly appreciate this thoughtful comment.

I think the manuscript is well written, English language needs of only minor editing.

Authors’ response: We hope that the minor language editing we have made to the manuscript will satisfy the reviewer’s concerns.

Round 2

Reviewer 1 Report

Comments and Suggestions for Authors

Table 1.

The citation in Table 1 for the management for stage IA3 per ‘FIGO guideline [32]’ should be corrected as ‘NCCN guideline [33]

Reviewer 2 Report

Comments and Suggestions for Authors

Because this paper is a narrative review, I suggest to include within the references the only RCT on follow up regimen published on JCO on july 2020 in order to support the best follow up regimen to be suggested.
